# A Training Programme for Developing Social and Personal Resources and Its Effects on the Perceived Stress Level in Adults in Daily Life—Study Protocol for a Prospective Cohort Study

**DOI:** 10.3390/ijerph20010523

**Published:** 2022-12-28

**Authors:** Christoph Janka, Tanja Stamm, Georg Heinze, Thomas E. Dorner

**Affiliations:** 1Center for Public Health, Institute of Social and Preventive Medicine, Medical University of Vienna, Kinderspitalgasse 15/1, 1090 Vienna, Austria; 2Center for Medical Data Science, Institute of Outcomes Research, Medical University of Vienna, Spitalgasse 23, 1090 Vienna, Austria; 3Ludwig Boltzmann Institute for Arthritis and Rehabilitation, 1090 Vienna, Austria; 4Center for Medical Data Science, Institute of Clinical Biometrics, Medical University of Vienna, Spitalgasse 23, 1090 Vienna, Austria; 5Karl-Landsteiner Institute for Health Promotion Research, Haus der Barmherzigkeit—Clementinum, Paltram 12, 3062 Kirchstetten, Austria

**Keywords:** health, training, social resources, personal resources, wellbeing, health behaviour, perceived stress, social relationships, life goals, meaning in life

## Abstract

Persistent stress and insufficient coping strategies have negative consequences for physical and mental health. Teaching adults the skills needed to sustainably improve stress-buffering aspects of their character could contribute to the prevention of stress-related diseases. In this non-randomised, observational, prospective cohort study, participants of a training programme for developing social and personal skills, to which they previously self-assigned, are assessed. The 12-month training programme focuses on improving perceived stress level (primary outcome), health behaviour, presence of common somatic symptoms, satisfaction with life, quality of social relationships, and wellbeing by addressing life goals, meaning in life, sense of coherence, social and personal resources, and transcendence. Study participants are recruited from the training groups via the training organiser. Companions, persons with whom they share a close relationship, are recruited to assess the interpersonal diffusion effects of the training. Matched individuals not participating in the training are the control group. Parameter assessment follows a pre-, post-, and follow-up (6 months) design. Designed to improve health-related outcomes in adults by addressing personality characteristics and using Lozanov’s superlearning principles to improve learning efficiency, this training programme is, to the study team’s knowledge, the first of its kind. From a research perspective, the outcomes of this study can provide new insights into primary prevention of stress-related diseases and how the effects of these measures are passed on through common personal interaction. The trial has been pre-registered (registration number: NCT04165473).

## 1. Introduction

Stress not only means a burden for individuals, but it also puts a financial burden on economies. The burden for the Austrian economy in 2010 due to stress-related mental illnesses in the working population was EUR 7,000,000.000 [1]; the latest numbers for the European Union reached over EUR 600,000,000.000 [2], and numbers for the total population are, therefore, even higher.

People’s interactions with their surroundings mould them as they grow up in a community [3]. This is normal and necessary for human growth, and it may be viewed as a method for coping with the surrounding environment [4,5]. Resilience is found to be a key enabler to promote health and wellbeing [6]. How quickly individuals can recover from adversity and how well they can endure stress depend on their personality and their learned abilities and coping techniques throughout life [7,8]. Stressful life experiences have been demonstrated to have long-term harmful impacts on mental and physical health, which can last a lifetime in some situations [9,10]. Daily stress has been demonstrated to affect the body at the cellular level, leading to the emergence of age-related illnesses earlier in life [11], and to have an effect on the immune system, making the person more susceptible to infections [12]. As a result, there is a strong link between perceived stress and an individual’s health state [13]. According to the World Health Organisation [14], one of the main skills required for optimal mental health is the capacity to cope with daily life stressors.

In the past, many stress management interventions under investigation, including psychoeducation, relaxation training, cognitive behavioural therapy, social support, coping skills training, or mindfulness training, have shown effects on reducing perceived stress and anxiety in graduate and undergraduate students [15]. While these measures are mainly focusing on reducing the stress response to build resilience, interventions following positive psychology principles can be considered to be more holistic as they embrace the past, the present, and the future on an individual as well as on a group level [16]. Many aspects and their positive effects on wellbeing, resilience, health, and happiness have been described [17]. Nevertheless, no intervention or training so far has been designed to not only teach and practice new skills, but also to support the participants in mastering challenges in the learning and implementation process.

The teaching content together with the teaching methodology may hold the key for students to acquire new skills better and more sustainably. To this day, most of the research on accelerated teaching effectiveness is conducted with children or teenagers. There is only a limited number of studies on adult teaching using accelerated teaching methods [18] and no studies conducted on acquiring the skills and their relations on the outcomes proposed in this research.

Through the curriculum of the training program under research, participants learn the skills through exercises that are designed to address all categories of learning, and new behaviours can be applied in real life. It is also designed in a way to cover the past, present, and future life of each participant. As social skills cover a great part of the programme, a diffusion effect to persons having a close relationship to the participants is expected and is assessed for.

A training programme could hold the chance for higher sustainability of its effects, if it provides improvements for mind, body, and the social aspects in a person’s life. Established concepts have been selected and assigned to each of these aspects. As shown in Figure 1 for the “mind”-aspect life goals, meaning in life, satisfaction with life, sense of coherence, transcendence, and wellbeing have been selected. For the “body”-aspect, perceived stress, health behaviour, and presence of common somatic symptoms have been selected, and for the “society”-aspect, quality of social relationships and social and personal resources have been selected.

A description for each concept based on its relevance and the dependency to other concepts is given in Table 1.

Feasibility results of the pilot study helped to design and develop the current study. They were mainly used for sample size calculations and optimising the assessment procedure. If a training programme can alter the scores on the scales for these concepts and what the interrelatedness looks like is subject of the planned study.

### Study Objectives

The major aim of this study is to look at the impact of a psychosocial training program for social connections on individuals’ reported stress levels in everyday life. Furthermore, the impacts of the training on metrics such as health behaviour, the existence of common somatic symptoms, life satisfaction, the quality of social interactions, and increases in wellbeing are being explored. The relevance of the participants’ life goals, meaning in life, social and personal resources, and experience of transcendence and feeling of coherence are also evaluated factors that are expected to be altered by the training intervention, henceforth referred to as the “training aims”. The secondary aim is to compare the intervention group’s outcomes to those of a pairwise-matched control group in order to assess the genuine effect of the intervention and its propagation to a closely related individual.

It is a novelty to evaluate a training programme with this theoretical background. If the principles behind the development of the training aims can be corroborated, it indicates the possibility of growing personal resources at the adult level. Figure 2 shows the concept of “training aims” and “outcome parameters” with “training aims” being in the centre, indicating its possible underlying nature.

Furthermore, this research might lead to a better knowledge of how to maintain and enhance one’s health, as well as provide insight into good functioning and the role of social ties in the primary prevention of stress-related disorders. The experiences of this study will enable further research to generate a more detailed understanding of measures for the prevention or treatment of stress-related illnesses.

## 2. Materials and Methods

### 2.1. Theoretical Background of the Training

The training programme in this study uses Lozanov’s superlearning principles to improve the knowledge and skills of participants by creating a relaxing and supportive atmosphere that enhances the learning process [30], structures the training exercises following a specific protocol for best integration, and applies presentation methods that appeal to visual, auditory, and tactile perceptions [31]. Within this setting, the students experience positive emotions, which has shown to be, among others, positively related to academic achievement [32] and contributed to students’ positive teaching experiences [33]. Additionally, negative emotions, such as fear of failure, are reduced, which has been shown to be a predictor for academic performance as well [34]. The trainers know how to provide the right setting for the students, with having high expectations for a positive outcome being one of them [35]. The positive or negative effects that teacher expectations, both high and low, can have on a student’s performance has been indicated in many studies for over five decades [36].

#### 2.1.1. Logical Categories of Learning

This concept is based on Gregory Bateson’s theory of the learning process [37]. It was used in the discipline of personal character development by Robert Dilts [38]. For the needs of this training, Kutschera further refined this concept and named it logical categories of learning [39,40] (see Figure 3). The levels below are influenced more strongly the higher the level at which individuals learn. During the training, all these levels will be addressed.

#### 2.1.2. Five-Roles Model

The Five-Roles model is a theory based on Kutschera [41]. Fulfilling one’s needs in each of the five roles (see Figure 4) “at work” and “in private life” is seen as a prerequisite for living a fulfilling life as an individual as well as in the community. In the training, the participants are guided to define how they aim to live each of the roles in their daily life.

### 2.2. Study Design

The planned cohort research, which evaluates a training program with training locations in Austria, Germany, and Switzerland, has a non-randomised, longitudinal, prospective design. The training program is commercially available on the market; hence, a study design without randomisation was used to address the research objectives without interfering with the training provider’s business practices. On the first day of the training programme, the individuals are recruited to participate in this study. For the further recruiting, a snowball sampling technique is used. Training participants will, therefore, receive a folder with detailed instructions on the assessment and on how to forward the other questionnaires and informed consent sheets contained in the folder. The control group is recruited via training participants’ companions to avoid direct contact between the control and intervention group. The control group is matched with the intervention group by gender, age (±5 years), employment status (employed full-time/employed part-time/unemployed/in school/on educational leave/on maternity or paternity leave/retired), family status (child or children/no children), relationship status (single/in a permanent relationship/undecided), perceived stress (±3 points on the Perceived Stress Scale (PSS)-10 scale), daily stressors (±30 points on a daily hassles questionnaire) and stressful life events within the last 3 months (yes/no). If a suitable sample size cannot be attained, the matching is conducted in steps, starting with 7 matching criteria and down to 4 criteria. Members of the control group who have fewer than four matching attributes are not included in the research. To guarantee a large enough matched control group, the training participant’s companion is asked to recruit two individuals that fit the matching criteria as well as possible.

The follow-up assessment is 6 months after the intervention has ended. To reach the estimated sample size, multiple training groups are merged. Tests with paper and pencil are used to assess the participants in the intervention and control groups, as well as their companions.

An overview of the study design is provided in the CONSORT flow diagram in Figure 5.

The criteria participants must meet to be eligible for this study are shown below. Those taking the training course do have to meet additional criteria, which are also stated in Table 2.

Figure 6 shows the timeline of the study, the duration of the training programme, the training modules, and the assessment points. The six training modules take place around every 8–10 weeks (at T_0_–T_5_), totalling in a one-year duration for the whole training programme. The study protocol follows the SPIRIT 2013 guidelines (see Appendix A).

### 2.3. Outcome Measures

#### 2.3.1. Primary Outcome

The PSS score absolute change between T0 (baseline) and T5 (post-intervention), as well as between T0 and T6 (follow-up), is the primary outcome.

#### 2.3.2. Secondary Outcomes

The absolute changes in presence of common somatic symptoms, in health behaviour, in satisfaction with life, in the quality of relationships, and in wellbeing between T_0_ and T_5_ and between T_0_ and T_6_ are the secondary outcomes. Self-report paper–pencil examinations are used for all assessments.

#### 2.3.3. Measurements

Every involved person is evaluated at three points in time: before the intervention (T_0_), at the end of the intervention (T_5_), and 6 months after the intervention has ended as a follow-up (T_6_). At T_0_ and T_5_, participants in the training programme are given the assessment forms in the training class. In a prepared folder, the training participants also receive the other forms—for the person with whom they have a close relationship, for their matched control, and for the matched control’s close-relationship person. At T_6_, all participants receive their forms via postal delivery, and a reply envelope is included.

All items are assessed pre-intervention (on the first day of the training programme) and post-intervention (on the first day of the final training module) as well as 6 months after the final module as a follow-up, except for the “expectations for the training programme”, the participants’ “intention to complete the whole training programme” (both only at T_0_), and the “frequency of application of trained content” (only at T_6_). Table 3 and Table 4 describe the tools and assessment items used in more detail.

The German versions of the surveys are utilised when they are available. If a German version of an assessment instrument is not available, a forward–back translation [54] as well as cultural modification of the questions in the form of a cognitive debriefing are performed to develop a German version of the instrument.

### 2.4. Sample Size

Using a two-group *t*-test with a 0.05 two-sided significance level, a sample size of 43 in each group has 90% power to detect a change in the mean improvement on the PSS of 5 points from T_0_ to T_6_—the mean improvement in the intervention group is assumed to be 10 and the difference between the mean improvement in the control group is assumed to be 5. Based on previous pilot data the common standard deviation is assumed to be 7.

Because of the quasi-experimental design, we expect the overlap in patient characteristics to be imperfect, requiring residual confounding to be adjusted for, resulting in an effective sample size of 50–60% of the original sample. Furthermore, we estimate a 20% drop-out rate. As a result, the sample size will be set at 100 per group. The recruitment procedure will be extended until the sample size is met.

### 2.5. Intervention

The study proposed attempts to answer the research questions by assessing participants of an existing training programme:

The trainings are held by Institut Kutschera GmbH in Austria, Germany, and Switzerland, and the programme is called “Resonanz^®^ Practitioner” [55]. The Institute is accredited by ISO 9001 [56], the course has been nationally approved, and Steinbeis University Berlin has given its academic accreditation for the course’s curriculum. The content equals 528 working hours [57].

The instructors meet all legal standards and have at least ten years of experience teaching this curriculum. Trainees, varying in number, are participating under the guidance of the trainers.

The intervention consists of six training modules (4 × 3 days, 2 × 5 days) and lasts approximately one year. Modules 1–3 start on a Thursday afternoon at 5 p.m. and end on Saturday at approximately 3 p.m., and modules 4 and 6 start on Tuesday at 5 p.m. and end on Saturday at approximately 3 p.m. The participants join each module in person. The group size ranges from 10 to 35 individuals. Each module has a specific topic and consists of a theory section and a tailored combination of exercises where the participants experience the different communication skills as individuals or in group settings. The titles and the content of the modules are presented in Table 5.

In addition, 5 units (50 min each) of self-experience are featured, in which participants have a private session with one of the trainers to explore a relevant topic. Ten protocols (one page each, following a predetermined framework) must be created in which participants record their impressions throughout a conversation to assist them to reflect on their previously acquired abilities. This is discussed under supervision. One book presentation is required to demonstrate that the participants engaged with a self-selected topic relevant to their resource portfolio and personal growth. Additionally, 50 h of peer-group sessions need to be held amongst the training participants to practise. The documentation of adherence to the training programme is conducted by the provider of the training programme for each individual for administrative purposes and is shared with the investigators for the purpose of this study.

#### 2.5.1. Intervention Group

The intervention group members participate in the course modules, as depicted in Table 4. The lecture materials are provided to them in a book and scriptum. The research does not pay any of the participants’ training expenses. Prior to or as soon as possible on the first day of the first and sixth modules, as well as six months after the training is through, the assessment is conducted in paper-and-pencil style. The training participants who also take part in the study are compensated. The participants will receive a compensation of EUR 30 for completing and submitting all 3 questionnaires. Dropouts will be compensated proportionately (EUR 10 for each completed questionnaire).

#### 2.5.2. Control Group

Both the intervention group participants and the control group participants undergo assessments at the same time. In Figure 6, the assessment timepoints are presented. The control group receives no treatment. All participants receive EUR 30 in reward for completing all three questions. Drop-outs from the research will be compensated proportionately (EUR 10 for each completed questionnaire). The control group is evaluated using paper-and-pencil questionnaires and is handled as a parallel group up to the completion of the intervention.

### 2.6. Intervention Fidelity

Two certified trainers will deliver the training protocol. Trainees are instructed to add changes to a printed version of the training schedule and hand it over to the principal investigator of this study after each training module. Deviations from the intended protocol might occur based on the needs of the participants in the groups. Concordance rates are reported for each 12-month training course.

### 2.7. Data and Statistical Analysis

IBM^®^ SPSS^®^ Statistics for Mac, Version 27 software (IBM Corp., Armonk, NY, USA) is used for all the statistical analyses. Initially, data are screened for plausibility using descriptive statistical analysis, and implausible values are cross-checked with the original participant records. To assess the influence of the intervention on the primary and secondary outcome variables (a) and to assess the mediating effect of the training items, labelled as “training aims” (b), a multivariate analysis of covariance (MANCOVA) is used. For both setups, a separate model is developed. The baseline PSS values for the relevant outcome variables and the matching criteria (see “Trial design”) are considered as covariates in these analyses, with the intervention group serving as the main factor variable.

The “training aims” items are included as variables in the mediation analysis (b), and their partial R^2^ values is used to assess their relevance in explaining the outcome variable. Furthermore, the mediation analysis determines how much of the intervention effect is mediated by the “training aims” items. This analysis is carried out for all primary and secondary outcomes, which include changes from baseline to the completion of the intervention as well as changes from baseline to the end of follow-up.

The companion group of the intervention group is compared to the companion group of the control group using the same analytic approach. Scatter plots and correlation coefficients are used to analyse the relationship between changes in primary and secondary outcomes assessed in trial participants and their closely related individuals. The research, as outlined, examines one primary endpoint and numerous secondary endpoints. Since the findings for the secondary endpoints are all to be presented, no multiplicity correction is undertaken. An intention-to-treat analysis is used for the primary endpoint improvement of the PSS score.

As a secondary endpoint, the PSS score is analysed using the per-protocol principle (all training modules completed). Missing items in assessments are imputed for the intention-to-treat analysis by carrying forward the preceding assessment, except at baseline, where closest-neighbour imputation is utilised. Imputation is not performed for the per-protocol analysis. A complete case analysis is undertaken as a sensitivity analysis.

### 2.8. Ethics Approval and Dissemination

In accordance with the Helsinki Declaration, all subjects are participating in the study on a voluntary basis. Before entering the study, the project administration obtains written informed consent from each participant. The study was authorised by the Medical University Vienna’s local ethical committee (Ref: 1592/2019) with the date of 25 October 2019. The protocol was additionally registered at clinicaltrials.gov (Identifier: NCT04165473). The Research Ethics Committee shall be notified of any necessary protocol changes together with the given reference number. The Clinical Trials register will be updated with these changes.

Participants’ personal information obtained throughout the consent and data collecting procedures is securely kept in a locked cabinet in a closed office. For the objectives of this study, a data monitoring committee is not required because the overall risk of harm is minimal. Only the researchers have access to the original, uncoded data.

## 3. Discussion

Perceived stress was chosen as a primary outcome for several reasons. Perceived stress can be seen as a sum parameter for individual imbalances. Individual, as one person might perceive the same situations in life differently than another person. Therefore, each person might need a different approach when it comes to stress mitigating measures. This is, on the one hand, a challenge, but on the other hand, it is a chance to find ways to address underlying mechanisms that work sustainably, without being practiced on a regular basis, for a greater range of individuals. Stress is also something, when it comes to self-assessment, that people can more easily relate to. Additionally, in future research, clinical parameters related to stress can be used to study the results and mechanisms in more detail.

Previous interventions on adults very often happened in a clinical setting and not, like this study, in a non-clinical setting. Studies with adults in the general population and health promotion can be found in the field of positive psychology in an organisational setting, where positive effects on job-related parameters and wellbeing were found. Stress as an outcome parameter only played a minor role and so did social relationships [58,59]. Studies on Loving Kindness Meditation, a form of meditation that fosters the experience of positive emotions, has shown positive effects on health and social relationships in adults. With continued meditation, it showed the most sustainable effect [60]. The type of intervention might be different to this study as it only focuses on meditation, but the intention and the outcome parameters are similar. There was no assessment for a diffusion effect. The same can be said for Mindfulness-Based Stress Reduction, which has been very well studied and for which evidence of its effectiveness on many health-related outcome parameters is given [61]. A study with a modularised training programme for older adults showed a reduction in the number of depressive symptoms and increased levels of life satisfaction, gratitude, and happiness [62]. However, as it was specifically designed for older adults, it cannot be used for a more representative sample of the population, whereas the study under research aims for that and embraces a multigenerational set of participants. Positive psychology interventions on social relationships have shown improvements in relationship satisfaction [63], although the follow-up period of 6 weeks was short and only the participants, without their counterparts in their relationship, were assessed. None of the trainings in the studies found used suggestopedia as a teaching method, used a concept founded on the categories of learning, and had the amount of training hours as the intervention under research. Additionally, the mind–body–society interaction was not so much of a focus as in this study, nor has a possible diffusion effect been assessed for. This study aims to merge several effective measures into a new, maybe more advanced, concept.

The fact that this study has a non-randomised controlled design is a limitation. On the other hand, long-term events naturally occurring in the field are difficult to randomise without influencing the process itself. In this case, this was the only design possible to answer the research questions and to not interfere with the training provider’s business practices. Additionally, matching, different training groups, different starting points for the training programmes, and a large sample size should overcome some of the drawbacks of the given design. A portion of the participants might join the programme as part of their employer’s educational programme or might be given the training as a present by a family member; others might only join a friend or partner.

The recruiting of the study participants comes with some limitations. On the one hand, there are the training participants that self-assign to the training—this will eventually cause a bias compared to a randomised controlled setup, although not all training participants will choose to participate by themselves; a good portion will take the training as part of an education programme of their employer, as a present from a friend or relative, or for private educational matters, without having the urge to reduce their perceived stress level.

As one of the inclusion criteria is not to having participated in a training before, the expectation and assumptions of the training is very similar amongst the individuals in the intervention group. The potential matches are provided with the link to the training programme on the homepage of the training provider to assure that they have the same information on the training as the intervention group. This should reduce the influence of the self-assignment to the intervention group compared to a randomised controlled setting. On the other hand, there are the training participants’ companions and the control group that are selectively recruited, meaning there should be a higher chance of meeting the matching criteria than by a recruitment in the general population. This should help reach the number of necessary participants faster as it should be more likely to meet the matching criteria and it should also be more likely that the recruited individuals participate in the study as they are recruited by a familiar person.

The instruments for the outcome parameters are validated, and the Perceived Stress Scale was selected as it has shown correlation with objective, diagnostic measures such as heart rate variability [64] or salivary cortisol concentrations [65], which are often used as stress indicators.

A recent study evaluating projects in Sweden on language acquisition (some of them funded by the European Union) agreed on the socio-cultural aspects being present in this teaching style but criticised the lack of teachers’ attention for the students’ experiences and the parent–child-like teacher–student relationship [66]. The training programme under research is having a Questions and Answer session included in each exercise to be able to consider special needs of the students and establish a professional adult–adult relationship throughout the training programme.

A part of the training program worth mentioning is its novelty when it comes to supporting its adult participants in the learning and implementation process. It is designed in a way that in every module, a specific exercise is included that explicitly addresses the participants’ blockages in the mentioned processes. The exercises are meant to help the participants overcome these hindrances and be able to continue in their process. The authors believe this is an essential aspect for the effectiveness and sustainability of a training program. These blockages can be limiting beliefs, unfavourable emotions, or hindering strategies. All exercises are shown in Appendix B.

There are some limitations associated with the implementation and scaling of the training program. To be able to train the content of the programme, the trainers themselves must take this training, as well as further education offered by the training provider, which, in total, makes a minimum of 3 years of experience with the training content and the teaching methodology. This is comparable to the length of the training to become a teacher in the public school system in most countries. While in some countries, the fees for the teachers’ education are almost fully covered by the government, only parts of the training programme under research are subsidised (in Austria, Germany, and Switzerland). This, on the one hand, might limit the speed of a worldwide scale-up of the programme, while on the other, it could bring more sustainability due to the knowledge and expertise the trainers have gained. The trainer education, together with its continued education programme, is designed to be approved to work as a trainer full-time, which also increases the sustainability of the scale-up of this programme. Over the last few years, the training provider has proven the scale-up concept with a successful implementation in Switzerland and Romania, where, at first, training programmes were held locally, and the graduates of the programmes became trainers themselves. A successful pilot project in Jordan has shown the validity of the training in cultures outside of Europe.

Another limitation comes with the accessibility of the training programme for the lower-income population due to the costs of the programme. Although many participants use the option to pay for the training in rates or their employer covers the costs as part of an employee education programme, financial constraints may pose a barrier for individuals with lower income.

The training aims parameters and outcome parameters were chosen based on the investigator’s understanding of a holistic health approach. Earlier research and pilot data indicate the impacts of the training programme, but only a fraction of this holistic idea has been considered. The approach to assessing interpersonal diffusion effects of an intervention is relatively new and could reveal new insights into social interaction and its relevance for healthy living. If this study is successful, it would show that a change in mindset, in particular life goals, meaning in life, sense of coherence, social and personal resources, and transcendence can be achieved sustainably in adults and that this has positive health effects. This should make further, more detailed research possible, to find out more about the changeability of these aspects during adulthood. Additionally, the importance of social relationships is highlighted, and it is shown that the quality of social relationships can be improved by addressing underlying skills. The standardised curriculum of the training programme would make a transfer to health institutions feasible. In further research, these underlying skills and found relationships can be investigated in more detail.

## 4. Conclusions

This project provides insights into the interplay of mind, body, and the social environment and how to influence it sustainably in a beneficial way. From measures to maintain health to supportive treatment for severe health conditions, these results could be integrated into practical applications. The study adds value to the understanding of what holistic health can look like and whether a training programme, such as the one under research, can explain how aspects that contribute to holistic health can be addressed. Findings from the proposed study are expected to contribute to the knowledge base of health professionals from different areas.

Previous studies have touched on parts of the parameters evaluated in this study. Aspects such as training aims and their relation to the outcome parameters, diffusion effects to individuals to which training participants have a close relationship, a concept based on the categories of learning and using suggestopedia that has shown its benefits in other areas of learning, outline the novelty of this study. Exercises that help participants solve blockages during the learning process could increase the effectiveness of the intervention and its sustainability.

A large portion of today’s population could benefit from stress mitigation interventions such as the one under research in this proposed study. It would also mean an alleviation of the financial burden of the effects of stress.

## Figures and Tables

**Figure 1 ijerph-20-00523-f001:**
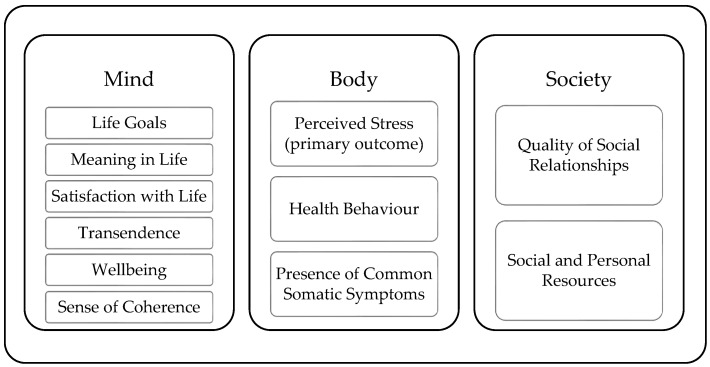
Idea for a holistic understanding of a mind–body–society combination and the underlying concepts we planned to use.

**Figure 2 ijerph-20-00523-f002:**
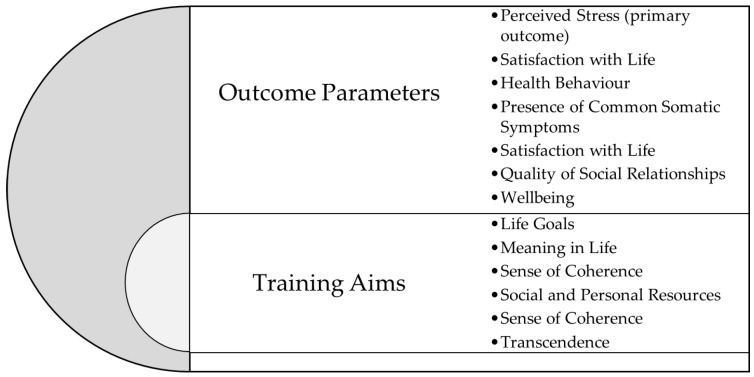
Training aims of the intervention and outcome parameters in this study.

**Figure 3 ijerph-20-00523-f003:**
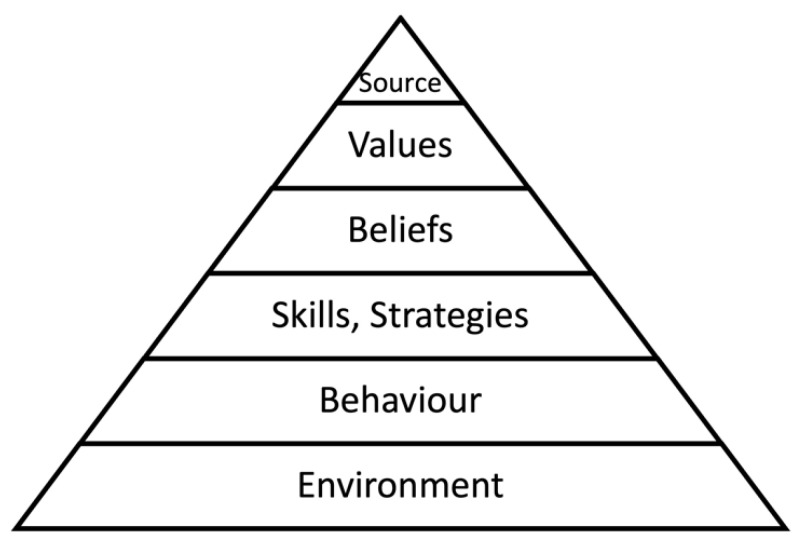
Logical categories of learning [41].

**Figure 4 ijerph-20-00523-f004:**
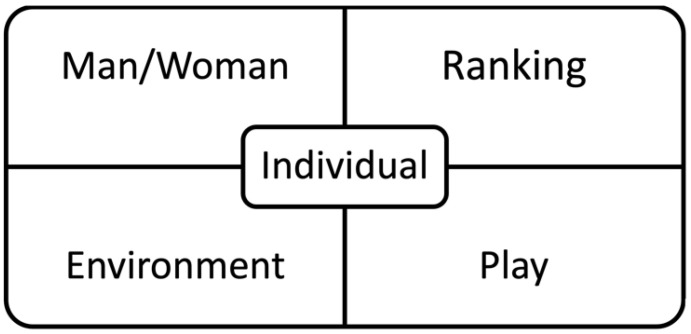
Five-Roles model of personality [41].

**Figure 5 ijerph-20-00523-f005:**
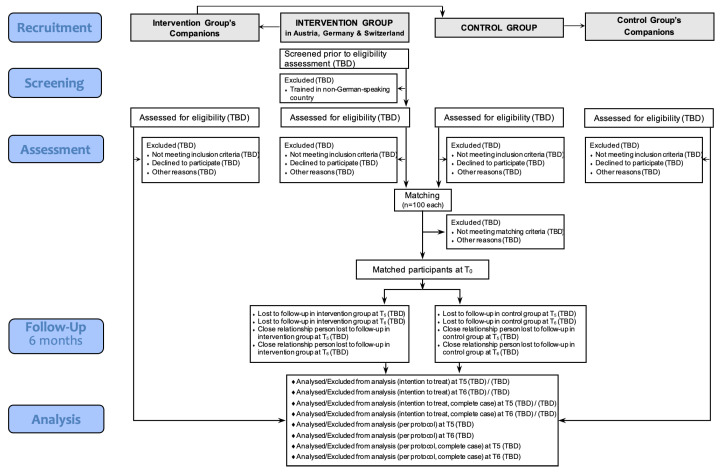
CONSORT 2010 Flow Diagram (Clinical Trial Registration Number: NCT04165473).

**Figure 6 ijerph-20-00523-f006:**
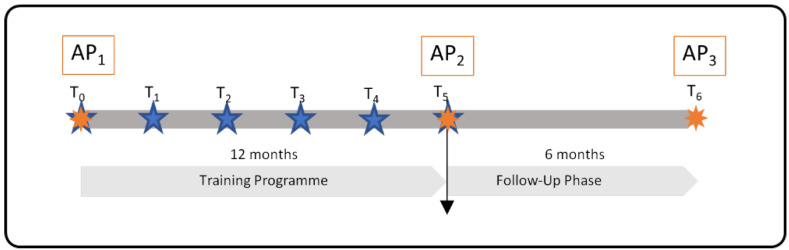
Training Timeline and Assessment Points. T_0–5_: 
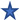
 Training Modules, AP_1–3_: 
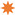
 Assessment Points.

**Table 1 ijerph-20-00523-t001:** Concepts we planned to use in this study and their descriptions.

Concepts	Description
Perceived stress	Perceived stress is a subjective measure of the degree to which a person feels overwhelmed or unable to cope with the demands of a situation. It is a key predictor of a range of negative health outcomes, including physical and mental health problems [19].
Health behaviour	Health behaviour refers to the actions that individuals take to maintain or improve their health. This includes engaging in healthy behaviours such as regular physical activity and a healthy diet, as well as avoiding unhealthy behaviours such as smoking and excessive alcohol consumption [20].
Presence of common somatic symptoms	Common somatic symptoms are physical complaints that are commonly experienced by individuals, such as headache, fatigue, and sleep disturbances. These symptoms can be influenced by a range of factors, including stress [21].
Satisfaction with life	Satisfaction with life refers to an individual’s overall evaluation of their life and their level of contentment with various aspects of their life. It is a key predictor of overall wellbeing and has been found to be negatively affected by high levels of perceived stress [22].
Quality of social relationships	Quality of social relationships refers to the positive and negative interactions that individuals have with others, as well as the overall quality of their relationships. Strong social relationships are associated with better mental and physical health outcomes, while poor social relationships can have negative effects on health and wellbeing [23].
Wellbeing	Wellbeing is a broad concept that refers to an individual’s overall sense of physical, mental, and emotional health and happiness. It is influenced by a range of factors, including perceived stress, social relationships, and health behaviours [24].
Life goals	Life goals are the long-term aspirations that individuals have for their lives. They can include career goals, family goals, and personal goals and can play a role in an individual’s overall sense of meaning and purpose in life [25].
Meaning in life	Meaning in life refers to an individual’s sense of purpose and the goals and values that give their life meaning. It has been found to be a key predictor of overall wellbeing and has been found to be negatively affected by high levels of perceived stress [26].
Sense of coherence	Sense of coherence refers to an individual’s ability to make sense of their life and the world around them. It is characterised by a sense of understanding, meaning, and predictability and has been found to be protective against the negative effects of perceived stress [27].
Social and personal resources	Social and personal resources refer to the internal and external resources that individuals have at their disposal to cope with stress and challenges. These can include things such as strong social support networks, an internal locus of control, and a problem-solving attitude [28].
Transcendence	Transcendence refers to the ability to find meaning and purpose beyond oneself. It has been found to be a protective factor against the negative effects of perceived stress and has been linked to overall wellbeing [29].

**Table 2 ijerph-20-00523-t002:** Criteria for including or excluding study participants.

Inclusion Criteria:	Exclusion Criteria:
Male, female, other Fluently German speaking Between 18 and 70 years of age Additional criteria for the intervention group: For the intervention group: Intention to complete the whole training or undecided	Score <19 on the Perceived Stress Scale Did not provide informed consent Additional criteria for the intervention group: For the intervention group: has not completed 2 or more modules of the training in total

**Table 3 ijerph-20-00523-t003:** Detailed description of the assessment tools.

Domain	Tool
Daily stressors	Evaluated using a list of the most frequent hassles and stressors encountered every day; created by Techniker Krankenkasse (37 items) [42].
Health behaviour	Evaluated using the FEG—Questionnaire for the Assessment of Health Behavior [43]. Covers the behaviours that are pertinent to this study: exercise, sleep, and psychological stress (43 items). The total sum is used for further calculations.
Life goals	Assessed by the Aspiration Index that evaluates internal and extrinsic ambitions, which can be viewed as life goals [44] (35 items). Further calculations are based on the value ratio between presence and searching.
Meaning in life	Assessed by the MLQ—Meaning in Life Questionnaire [45], which distinguishes between presence of meaning and searching for meaning (10 items).
Perceived stress	Assessed by the PSS-10—Perceived Stress Scale [46] (10 items).
Personal and social skills	Personal, societal, and structural resources are assessed by the ERI—Essener Ressourceninventar [47] (33 items). The total mean value is utilised for further calculations.
Presence of common somatic symptoms	Assessed by the FEG—Questionnaire for the Assessment of Health Behavior [43] (5 items). The total mean value is used for further calculations.
Quality of relationships	Assessed by EVOS—Evaluation of Social Systems [48] and EXIS—Experience of Social Systems questionnaires [49] (68 items in total). The total sum is utilised for further calculations.
Satisfaction with life	Assessed by the LS-4—Life Satisfaction questionnaire [50] (4 items).
Sense of coherence	Assessed by the SOC-L9—Sense of Coherence Scale [51] (9 items).
Transcendence	Assessed by the CSRF—Character Strength and Rating Form (Transcendence part only) [52] (5 items).
Wellbeing	Assessed by the WHO-5 (by World Health Organisation) questionnaire for psychological wellbeing [53].

**Table 4 ijerph-20-00523-t004:** Detailed description of the additional assessment items.

Domain	Assessment Items
Coverage of training costs (self, other)	Single question developed by the researcher to assess the impact of cost coverage on the training effect (1 item).
Desire to improve social relationships	Single question developed by the researcher to assess the effects of the desire to strengthen social interactions on the training effect and to classify it as a confounder when there is a difference between the intervention and control groups. “Yes”, “sometimes”, and “no, I’m totally satisfied” are the response possibilities (1 item).
Expectations for the training programme	Assessed by a table where participants rate the effect they believe a training programme can have on the outcome parameters and the training aims.
Feeling of belonging to a group	Single question developed by the researcher to assess the impact of group identification on the training effect and to categorise it as a confounder when there is a difference between the intervention and control groups (1 item).
Frequency of application of trained content	Single question formulated by the researcher to evaluate the influence on the sustainability of the training effect. There are 6 response options representing increasing frequency (1 item).
Intention to complete the whole training programme	Single question developed by the researcher to assess the participants’ intention to complete the full training. “Yes”, “not sure yet”, “definitely not”, and “I have already attended other modules” are the response possibilities (1 item).
Medication	A single question is asked about the number of different medications that are taken.
Socio-demographic data	Age, gender, education, income, family status, relationship status, employment status, and earlier trainings (14 items).
Stressful life events within the last 3 months	Single question developed by the researcher to assess the impact of a stressful life experience that is still viewed as unpleasant on the training effect and to serve as a matching criterion (1 item).

**Table 5 ijerph-20-00523-t005:** Duration and titles of the training modules.

Module	Title	Duration
Module 1	Finding meaning and developing your personality	3 days
Module 2	Improving quality of life with others	3 days
Module 3	Solving conflicts respectfully	3 days
Module 4	Finding meaning and discovering your own strengths and weaknesses in a group	5 days
Module 5	Systemic understanding of change	3 days
Module 6	Summary and integration	5 days

## Data Availability

Data sharing is not applicable to this article. The trial is registered as a clinical trial on clinicaltrials.gov (ID number: NCT04165473, accessed on 20 September 2022).

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
