# Peer review of "A Training Programme for Developing Social and Personal Resources and Its Effects on the Perceived Stress Level in Adults in Daily Life—Study Protocol for a Prospective Cohort Study"

_ijerph, 2022, doi:10.3390/ijerph20010523_

Round 1

Reviewer 1 Report

The manuscript no doubt contributed significantly to the field of the psychosocial training programme for social relationships, the perceived stress level in daily life in adults and health behaviour. The title of the study is apt and attractive. However, the following are some of the concerns noticed in the manuscript:

1.   The plagiarism check was carried out and the similarity result of 34% is a bit high. The report is attached. Some of the areas that recorded high plagiarism include the second paragraph of the introduction, study objective section, 2.1.1, 2.2, 2.6, 2.9.1, 2.9.2. 2.11, 2.12, and 2.13 need to be paraphrased.

2.   The writing style needs to change. For example, it was discovered that authors used future tenses. Since the study has been concluded, the tenses should change.

3.   The use of first-person plural pronouns like we, us etc. should be avoided in this type of study. Check page 3, line 131, page 4, line 163 among others.

4.   Lines 357, 358, 368. 319, 400, and 401 require attention to correct all errors.

5.   The discussion section needs to be improved and supported with more existing similar studies as established in the literature on the subject matter.

Reviewer 3 Report

The issues raised are very important. Due to the fact that this study is of a scientific nature, it is necessary to operationalize the key concepts (in the first part of the article). The authors use a large number of terms such as well-being, stress, and promoting health. etc. In fact, the article does not show what it brings to the theory. In the case of scientific articles, it is important and necessary. There is also no reference to other tools. The authors did not discuss and the conclusions are very poor.

Reviewer 4 Report

This study protocol is unprecedented and is regarded for its novelty. The manuscript is appropriately described under SPIRIT. The non-randomized controlled design is a limitation in elucidating causal relationships, but the limits are overcome by matching and a large sample size.

p.2 ln. 86-89

“no intervention or training so far is designed to not only teach and practice new skills, but also to support the participants in mastering challenges in the learning and implementation process.”

Regarding the study novelty, what difficulties have prevented such training, and how have the authors overcome these difficulties? I would appreciate a supplementary explanation.

Round 2

Reviewer 3 Report

The article raises an important and current issue. However, it requires some corrections and additions.

While the research part is described in detail, which is probably due to the great personal involvement in the described research. It is the theoretical aspects that require significant improvement.

The introduction is quite long and of a very general nature.

Typically, scientific articles should define key concepts and possibly dependencies if they are mentioned in the text. Of course, based on available research and analysis. The title of the article contains many terms. This should be embedded in theoretical considerations. This article lacks it. In addition, the authors did not write what the conclusions of the research bring to practice and theory. Nothing was written about the limitations of the study.
